evolution, genomics, palaeontology

archaeology, geometric morphometrics, ancient DNA, migration, *Canis lupus familiaris*, circumpolar

**Author for correspondence:**
Carly Ameen
e-mail: c.ameen@exeter.ac.uk

†C.A., T.R.F., S.K.B. and A.L. contributed equally to this study.

# Specialized sledge dogs accompanied Inuit dispersal across the North American Arctic

Carly Ameen[1,2,†], Tatiana R. Feuerborn[3,4,6,7,8,†], Sarah K. Brown[9,11,12,†], Anna Linderholm[13,14,†], Ardern Hulme-Beaman[2,14,17], Ophélie Lebrasseur[2,14,18], Mikkel-Holger S. Sinding[5,6,20,23], Zachary T. Lounsberry[11], Audrey T. Lin[14,16], Martin Appelt[24], Lutz Bachmann[20], Matthew Betts[25,26], Kate Britton[27,28], John Darwent[9], Rune Dietz[29,31], Merete Fredholm[19], Shyam Gopalakrishnan[4,5], Olga I. Goriunova[32], Bjarne Grønnow[24], James Haile[14], Jón Hallsteinn Hallsson[33], Ramona Harrison[34], Mads Peter Heide-Jørgensen[35], Rick Knecht[27], Robert J. Losey[36], Edouard Masson-MacLean[27], Thomas H. McGovern[37,38], Ellen McManus-Fry[27], Morten Meldgaard[4,6], Åslaug Midtdal[39], Madonna L. Moss[40], Iurii G. Nikitin[41], Tatiana Nomokonova[42], Albína Hulda Pálsdóttir[21,33], Angela Perri[43], Aleksandr N. Popov[41], Lisa Rankin[44], Joshua D. Reuther[45], Mikhail Sablin[46], Anne Lisbeth Schmidt[24], Scott Shirar[45], Konrad Smiarowski[38,47], Christian Sonne[29,30,48], Mary C. Stiner[49], Mitya Vasyukov[50], Catherine F. West[51], Gro Birgit Ween[22], Sanne Eline Wennerberg[52], Øystein Wiig[20], James Woollett[53], Love Dalén[7,8], Anders J. Hansen[4,6], M. Thomas P. Gilbert[5,54], Benjamin N. Sacks[10,11], Laurent Frantz[55], Greger Larson[14,15], Keith Dobney[2,27,56], Christyann M. Darwent[9] and Allowen Evin[57]

[1]Department of Archaeology, University of Exeter, Exeter, Devon, UK
[2]Department of Archaeology, Classics and Egyptology, University of Liverpool, Liverpool, Merseyside, UK
[3]Department of Archaeology and Classical Studies, Stockholm University, Stockholm, Sweden
[4]Centre for GeoGenetics, and [5]Section for Evolutionary Genomics, GLOBE Institute, University of Copenhagen, Copenhagen, Denmark
[6]The Qimmeq Project, University of Greenland, Nuussuaq, Greenland
[7]Department of Bioinformatics and Genetics, Swedish Museum of Natural History, Stockholm, Sweden
[8]Centre for Palaeogenetics, Stockholm, Sweden
[9]Department of Anthropology, [10]Department of Population Health and Reproduction, and [11]Mammalian Ecology and Conservation Unit of the Veterinary Genetics Laboratory, University of California Davis, Davis, CA, USA
[12]Washington Department of Fish and Wildlife, Olympia, WA, USA
[13]Department of Anthropology, Texas A&M University, College Station, TX, USA
[14]The Palaeogenomics and Bio-archaeology Research Network, Research Laboratory for Archaeology and History of Art, [15]School of Archaeology, and [16]Department of Zoology, University of Oxford, Oxford, UK
[17]School of Natural Sciences and Psychology, Liverpool John Moores University, Liverpool, UK
[18]GCRF One Health Regional Network for the Horn of Africa (HORN) Project, Institute of Infection and Global Health, Liverpool, UK
[19]Department of Veterinary and Animal Sciences, and [20]Natural History Museum, and [21]Centre for Ecological and Evolutionary Synthesis (CEES) Department of Biosciences, University of Oslo, Oslo, Norway
[22]University of Oslo Museum of Cultural History, Oslo, Norway
[23]Greenland Institute of Natural Resources, Nuuk, Greenland
[24]National Museum of Denmark, Copenhagen, Denmark
[25]Canadian Museum of History, Gatineau, Quebec, Canada
[26]Department of Anthropology, University of New Brunswick, Fredericton, New Brunswick, Canada
[27]Department of Archaeology, University of Aberdeen, Aberdeen, UK

[28]Department of Human Evolution, Max Planck Institute for Evolutionary Anthropology, Leipzig, Sachsen, Germany

[29]Arctic Research Centre, and [30]Department of Bioscience, Aarhus Universitet, Aarhus, Denmark

[31]Department of Bioscience Roskilde, Aarhus Universitet, Roskilde, Denmark

[32]Laboratory of Archaeology and Paleoecology of the Institute of Archaeology and Ethnography (Siberian Branch of Russian Academy of Science), Irkutsk, Russian Federation

[33]Faculty of Agricultural and Environmental Sciences, The Agricultural University of Iceland, Reykjavik, Iceland

[34]Department of Archaeology, History, Cultural Studies, and Religion, University of Bergen, Bergen, Hordaland, Norway

[35]Birds and Mammals, Greenland Institute of Natural Resources, Copenhagen K, Denmark

[36]Department of Anthropology, University of Alberta, Edmonton, Alberta, Canada

[37]Department of Anthropology, Hunter College CUNY, New York, NY, USA

[38]The Graduate Center, City University of New York, New York, NY, USA

[39]Holmenkollen Ski Museum, Oslo, Norway

[40]Department of Anthropology, University of Oregon, Eugene, OR, USA

[41]Museum of Archaeology and Ethnography at the Institute of History, Archaeology and Ethnography of the Peoples of the Far East (Far Eastern Branch of Russian Academy of Science), Vladivostok, Russian Federation

[42]Department of Archaeology and Anthropology, University of Saskatchewan, Saskatoon, Saskatchewan, Canada

[43]Department of Archaeology, Durham University, Durham, UK

[44]Department of Archaeology, Memorial University of Newfoundland, St John's, Canada

[45]Department of Anthropology, University of Alaska Museum of the North, Fairbanks, AK, USA

[46]Zoological Institute of Russian Academy of Sciences, St Petersburg, Russian Federation

[47]Section for Cultural Heritage Management, Department of Cultural History, University Museum of Bergen, Bergen, Norway

[48]School of Forestry, Henan Agricultural University, Zhengzhou, China

[49]School of Anthropology, University of Arizona, Tucson, AZ, USA

[50]Department of Biological Diversity and Sustainable Use of Biological Resources, Russian Academy of Sciences, Moskow, Russian Federation

[51]Department of Anthropology and Archaeology Program, Boston University, Boston, MA, USA

[52]Government of Greenland, Veterinary and Food Authority, Nuuk, Greenland

[53]Département des Sciences Historiques, Université Laval, Quebec, Canada

[54]Norwegian University of Science and Technology, University Museum, Trondheim, Norway

[55]School of Biological and Chemical Sciences, Queen Mary University of London, London, UK

[56]Department of Archaeology, Simon Fraser University, Burnaby, British Columbia, Canada

[57]Institut des Sciences de l'Evolution—Montpellier, CNRS, Université de Montpellier, IRD, EPHE, Montpellier, Occitanie, France

(iD) CA, 0000-0002-4580-2125; AH-B, 0000-0001-8130-9648; OL, 0000-0003-0687-8538; ATL, 0000-0003-2505-1480; SG, 0000-0002-2004-6810; AE, 0000-0003-4515-1649

Domestic dogs have been central to life in the North American Arctic for millennia. The ancestors of the Inuit were the first to introduce the widespread usage of dog sledge transportation technology to the Americas, but whether the Inuit adopted local Palaeo-Inuit dogs or introduced a new dog population to the region remains unknown. To test these hypotheses, we generated mitochondrial DNA and geometric morphometric data of skull and dental elements from a total of 922 North American Arctic dogs and wolves spanning over 4500 years. Our analyses revealed that dogs from Inuit sites dating from 2000 BP possess morphological and genetic signatures that distinguish them from earlier Palaeo-Inuit dogs, and identified a novel mitochondrial clade in eastern Siberia and Alaska. The genetic legacy of these Inuit dogs survives today in modern Arctic sledge dogs despite phenotypic differences between archaeological and modern Arctic dogs. Together, our data reveal that Inuit dogs derive from a secondary pre-contact migration of dogs distinct from Palaeo-Inuit dogs, and probably aided the Inuit expansion across the North American Arctic beginning around 1000 BP.

# 1. Introduction

Dogs (*Canis lupus familiaris*) played a critical role in early human adaptation to circumpolar high-latitude environments. Early dog specimens from Late Pleistocene to Early Holocene sites throughout northeastern Asia [1–3], bear witness to this early association between humans and dogs in the Arctic. Recent genetic analyses indicate that the earliest dogs found in the Americas belonged to a now extinct lineage of Arctic dog that was introduced from Eurasia at least 10 000 years ago [4]. Aside from this initial peopling of the Americas, the North American Arctic has experienced additional human migration episodes of genetically distinct populations [5–10], which were accompanied by potentially distinct dog populations. The importance of dogs during these migrations, however, remains largely unknown [4,11]. Investigating whether or not these new groups brought genetically differentiated dog populations with them into the North American Arctic, and the relationship between these dogs and those already present in the region, is crucial for understanding the history of dogs in the Americas.

Archaeological evidence suggests that dogs were relatively rare in the North American Arctic prior to the Inuit period [12,13]. The Inuit emergence in Alaska beginning approximately 2000 BP brought large-scale changes in lifeways, subsistence practices and material culture to the North American Arctic. Their subsequent expansion translocated this culture out of Alaska eastward to Greenland, and along the coast of subarctic Eastern Canada starting in 1000 BP [14,15]. The rapid expansion of the Inuit is attributed in part to their exploitation of advanced transportation technologies, including the development and widespread usage of the umiak and kayak for sea travel, and the dog sledge for use on land and ice [16–18]. Despite the ubiquitous association of dog sledging with North American Arctic peoples, and the presence of dog sledge technology in Siberia by 9000 BP [1,19], there remains little firm evidence for dog sledging in the Americas before 1000 BP [12]. Today, sledge dogs remain culturally and economically crucial to indigenous lifeways in the Arctic, but dog numbers are declining rapidly due to changing climate, the recurrence of parvovirus and distemper, the preferential use of snowmobiles, and the culling of indigenous dogs during both historical and modern times [20,21]. Furthermore, the relationship between modern Arctic dogs and archaeological dogs from the Inuit and preceding Palaeo-Inuit periods remains unclear.

Previous studies have suggested that dogs associated with modern Arctic populations are the direct descendants of the populations that were brought by the Inuit [22–24], reaffirming the intrinsic technological role of dogs for modern Inuit who continue to occupy the North American Arctic. However, more recent introductions of dogs attributed to the European settlement of the Eastern Arctic beginning in the eighteenth century, and Alaskan settlers at various times in the nineteenth century, largely in connection to mining and settlement [22,23], probably also contributed to the genetic make-up of modern

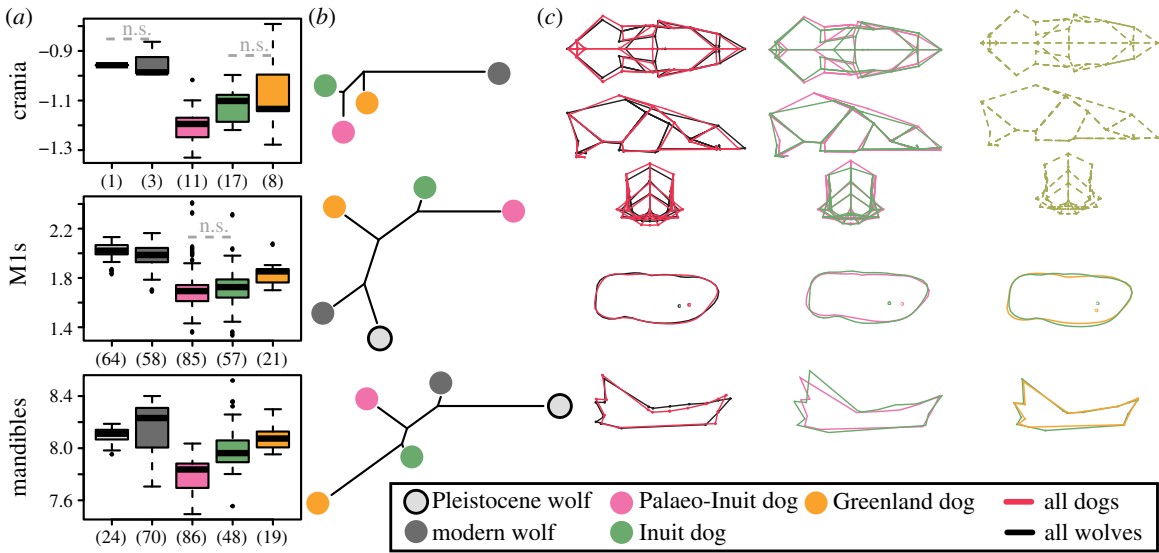

**Figure 1.** Morphometric variation of Arctic dogs and wolves. (*a*) Size variation of Pleistocene and Modern wolves, and Palaeo-Inuit, Inuit, historic and recent Greenland (Historic and modern Greenland breeds, see electronic supplementary material, text) dogs. Boxplot of the log-transformed centroid size with sample size shown in brackets. 'n.s.' highlight non-significant pairwise comparison (Wilcoxon's test) between neighboring groups (table 2). (*b*) Overall shape differentiation between groups shown as neighbour-joining networks derived from Mahalanobis distances for each element separately. (*c*) Visualization of the cranial (top), first lower molar (middle) and mandible (bottom) shape differences between: wolves (black) and all domestic dogs (red); Palaeo-Inuit dogs (pink) and Inuit dogs (green); and Greenland dogs (orange), and Inuit dogs (green). Shape differences are visualized along the discriminant axis between the groups. Wireframes with dashed lines indicate non-significant differences.

dog populations across the region. In order to disentangle the spatio-temporal patterns in past North American Arctic dog populations and their relationship to recent Arctic dogs, it is necessary to establish the morphological and genetic distinctiveness of these groups.

To do so, we examined the phenotypic changes associated with the arrival of Inuit dogs using geometric morphometric (GMM) analyses [25] to explore cranial, lower first molar and mandibular morphometric variation. We then analysed complete and partial mitochondrial genomes of pre-contact Arctic dogs alongside publicly available data for pre-contact dogs from across the Americas [4,26], as well as historical and modern Arctic sledge dogs [22,23]. Integrating these datasets allowed us to investigate the history of pre-contact dogs in the North American Arctic, contextualize the origins of Inuit dogs relative to earlier Palaeo-Inuit dogs, and clarify the relationship between these archaeological dogs and modern Arctic dog populations.

## 2. Results and discussion

### (a) The origins and legacy of pre-contact Arctic dogs

To better understand the diversity of ancient Arctic dog morphologies, we investigated the phenotypic variation between Palaeo-Inuit, Inuit and more recent Greenland dog (historical Arctic and modern Greenland breed, see electronic supplementary material, text), as well as Arctic wolf populations. To do so, we used GMM on the crania, lower first molars and hemi-mandibles (figure 1 and table 1; electronic supplementary material, figure S14). Overall, wolves were easily distinguishable from dogs (table 2). In particular, wolves were less morphologically diverse in lower M1 size and shape, and possessed consistently larger measurements, and narrower and lower braincases relative to our sampled dogs (figure 1).

Analyses of the dogs found that Inuit dogs differ from both Palaeo-Inuit and recent Greenland dogs (figure 1 and table 2). Inuit dogs tended to be larger than Palaeo-Inuit dogs, but

smaller than recent Greenland dogs with no allometric repatterning (except for the mandible between Palaeo-Inuit and Inuit; figure 1*a* and table 2). Comparatively, Inuit dogs possess a proportionally narrower cranium, a less elevated braincase, a wider lower M1 and a more developed mandibular ascending ramus than Palaeo-Inuit dogs (figure 1*c*). Where the differences are significant, the mean cross-validation between Palaeo-Inuit and Inuit dogs ranges from 71.0% to 83.8% depending on the element (table 2).

We detected no differences in the crania between the Inuit and recent Greenland dogs (table 2). However, Inuit and Greenland dogs did differ in the size, shape and form of their lower M1 and mandible (mean cross-validation ranging from 63.5% to 85.0%), with no change in size-shape relationship (allometries) nor variance. Compared to historical and modern Arctic dogs, Inuit dogs showed a proportionally wider molar and a more convex bend to the mandibular body (figure 1*c*). Inuit dogs exhibit a similar cranial shape to the recent Arctic dogs while differing from the Palaeo-Inuit dogs, though the Inuit dogs differ equivalently from both Palaeo-Inuit and recent Arctic dogs in their molar and mandible (table 2).

Given that the GMM analyses suggested that the dogs associated with the different cultural groups were morphologically divergent, we assessed whether Palaeo-Inuit, Inuit and historical and modern Arctic dogs were also genetically distinct. We obtained mitochondrial genomes from 186 samples with a minimum average coverage of threefold to investigate the mitochondrial diversity of Arctic dogs through time and trace patterns of migration. An additional 40 samples were assigned to specific haplogroups by generating mitochondrial control region sequences using Sanger sequencing (table 1). Globally, modern and ancient domestic dogs group into four major (A–D), and two minor (E–F) mitochondrial clades [4,26,29]. Our phylogenetic analyses revealed that nearly all of the sampled dogs belonged to the mitochondrial A clade (electronic supplementary material, figure S4). We also identified four major subclades within the A clade: A1a, A2a, A1b and

**Table 1.** Number of samples generated or used in study. (a) Morphometric sample size: number of samples per group and per element analysed. (b) Genetic sample size: numbers of samples successfully sequenced per group and per type of sequencing/number attempted.

|  |  | Palaeo-Inuit | Inuit | historical | modern | modern wolf | Pleistocene wolf | total |
|---|---|---|---|---|---|---|---|---|
| **(a)** |  |  |  |  |  |  |  |  |
| GMM | crania | 11 | 17 | 6 | 2 | 3 | 1 | 40 |
|  | mandible | 86 | 48 | 5 | 14 | 70 | 24 | 247 |
|  | lower M1 | 85 | 57 | 6 | 14 | 58 | 64 | 284 |
|  | no. of spec.[a] | 124 | 92 | 12 | 16 | 70 | 77 | 391 |

|  |  | Palaeo-Inuit | Inuit | historical | modern | Siberin Holocene | other[b] | total |
|---|---|---|---|---|---|---|---|---|
| **(b)** |  |  |  |  |  |  |  |  |
| DNA | Sanger[c] | 23/87 | 128/261 | 20/51 | 9/14 | 13/30 | 0/36 | 193/479 |
|  | 3 × coverage[d] | 12/41 | 84/197 | 62/126 | 14/14 | 14/27 | 1/14 | 199/431 |
|  | 10 × coverage[d] | 2/41 | 28/197 | 37/126 | 14/14 | 3/27 | 0/14 | 94/431 |
|  | no. of spec.[a] | 92 | 289 | 147 | 24 | 40 | 36 | 628[a] |

[a]no. of spec.: number of specimens. Some specimens underwent Sanger and next-generation sequencing or multiple elements from an individual were included in the GMM analyses.
[b]Cultural affiliations including medieval and historical Iceland, Norse and unknown.
[c]D-loop haplotypes obtained via Sanger sequencing/samples sequenced with Sanger sequencing from specified culture.
[d]Mitogenomes obtained with indicated mean coverage/samples sequenced with next-generation sequencing from specified culture.

A2b, three of which had been previously identified in the Arctic (A1b, A2a [23] and A2b [4]) (figure 2). By contrast, most modern European dogs have haplotypes in the A1a clade.

All previously sampled pre-contact dogs from the Americas, including the earliest specimens known from the continent, exclusively carried A2b haplotypes [4,11] (figure 2). The uniformity of this clade in early contexts and the arrival of dogs carrying different haplotypes in later periods suggests that specific dog populations were associated with different cultural group. Thus, our data show that early human migration(s) (prior to 10 000 years ago) into the Americas were associated with dogs carrying only A2b haplotypes, while later migrations (after 5500 years ago) into the North American Arctic introduced dogs carrying haplotypes belonging to A1a, A1b and A2a subclades (figure 2).

Of the 92 specimens that were extracted and sequenced from Palaeo-Inuit contexts, only 12 possessed sufficient DNA to allow them to be assigned to haplogroups. Despite this sample size, these data represent the only currently available genetic information from Palaeo-Inuit dogs. While most Palaeo-Inuit dogs possessed A2b haplotypes (83.3%), two Palaeo-Inuit specimens from Alaska possessed A2a haplotypes (CK-H37-M1, SEL-33-0057b) suggesting that the A2a haplotype was present in North American Arctic prior to the Inuit period (figure 2). We found haplotypes immediately basal to this A2 subclade in Siberia dating to several thousand years prior to the Inuit colonization of the North American Arctic, suggesting that the appearance of these lineages on Palaeo-Inuit sites was the result of the Siberian ancestry of the dogs and people (electronic supplementary material, figure S4). Thus, though the Inuit were likely not responsible for the first appearance of this dog lineage in the Americas, they were responsible for the considerable geographical expansion of this lineage into the Eastern Arctic, where, during the Inuit period, they became the most common haplotypes across the entire North American Arctic (figure 2).

Temporal shifts in mitochondrial haplotype frequencies suggest a near-replacement of Palaeo-Inuit dogs in the North

American Arctic coinciding with the Inuit expansion from Siberia (figure 2 and electronic supplementary material, figure S11). Demographic analyses (as reconstructed from Bayesian skyline plots) show apparent founder effects coinciding with the timing of the Inuit expansion into the eastern around 1000 BP (electronic supplementary material, figure S9). The pattern of haplotype frequencies between locations and time periods provide strong evidence that Inuit migrants brought dogs with them from Siberia (figure 2; electronic supplementary material, figure S11). In particular, the frequencies of haplotypes from different subclades differed strongly between the Palaeo-Inuit and Inuit Arctic samples ($F_{ST} = 0.33$, $p < 0.001$), but were similar between Siberian and North American Inuit samples ($F_{ST} = 0.04$, $p > 0.10$; see electronic supplementary material, text and figure S11). These results are consistent with the results of previous human genetic analyses which have linked Inuit groups to both Siberian and earlier Alaskan populations [30,31].

Inuit and historical populations also possessed different haplotype frequencies ($F_{ST} = 0.33$, $p < 0.001$; figure 2). In particular, the A1a haplotypes increased over the past 300 years. A more recent expansion of subclades A1a and A1b compared to those of A2a was further supported by a network analysis and Bayesian skyline plots (see electronic supplementary material, text and figure S12). The European exploration of both Greenland and the Canadian Arctic during the nineteenth and twentieth centuries, and the nineteenth century Alaskan Gold Rush, increased the interaction between indigenous Arctic groups and Europeans, facilitating longer distance travel of both dogs and people [24]. This cultural mixing brought Eurasian dogs to the region in large numbers [22], probably contributing to the increase in A1a haplotype frequency. On the other hand, two instances of A1a subclade haplotypes from northern Alaska in the Inuit archaeological sample suggest the possibility of drift as a partial explanation for the increase in frequency of this subclade over the past 300 years. Lethal epidemics in indigenous dogs have also led to large-scale population turnover and replacement by European breeds in many regions [24,32]. While the

**Table 2.** Pairwise comparisons between dogs and wolves, Palaeo-Inuit and Inuit dogs, and Inuit and recent Greenland dogs. Differences are assessed using Wilcoxon's test for size, MANOVAs for shape and form and MANCOVAs for allometries. Differences in variances are tested following [27] for shape and Fligner–Killeen tests for size. Leave-one-out cross-validation percentages were obtained from 100 linear discriminant analyses based on balanced samples and dimensionality reduction [28] and are presented as the mean and 90.0% confidence interval. Shaded boxes are non-significant results after correction of the *p*-values for multiple comparisons.

| | | Dogs versus wolves | | Palaeo-Inuit dog versus Inuit dog | | Inuit dog versus Greenland dog | |
|---|---|---|---|---|---|---|---|
| | | comparison | cross-validation (%) | comparison | cross-validation (%) | comparison | cross-validation (%) |
| crania | size | $W = 137, p = 8 \times 10^{-4}$ | 91.2 (62.5–100%) | $W = 141, p = 0.02$ | 71% (63.6–77.3%) | $W = 75, p = 0.7$ | 38% (12.2–50%) |
| | shape | $F_{4,35} = 9.16, p = 3.58 \times 10^{-5}$ | 86.7% (50–100%) | $F_{2,25} = 6.3, p = 6 \times 10^{-3}$ | 75.5% (68.2–86.4%) | $F_{2,22} = 2.3, p = 0.1$ | 56.4% (43.7–68.7%) |
| | form | $F_{4,35} = 2.89, p = 0.036$ | 85.5% (50–100%) | $F_{3,24} = 3.807, p = 0.023$ | 76.2% (63.6–82%) | $F_{3,21} = 2.03, p = 0.14$ | 54.4% (37.5–75%) |
| | allometries | $F_{35,2} = 1.19, p = 0.56$ | | $F_{23,2} = 0.9374, p = 0.64$ | | $F_{20,2} = 1.163, p = 0.56$ | |
| | size variance | $x^2 = 1.5, p = 0.22$ | | $x^2 = 4 \times 10^{-4}, p = 0.98$ | | $x^2 = 0.49, p = 0.48$ | |
| | shape variance | $d_{var} = 0.0012, p = 0.10$ | | $d_{var} = 3 \times 10^{-4}, p = 0.57$ | | $d_{var} = 5 \times 10^{-5}, p = 0.91$ | |
| M1 | size | $W = 18\,576, p < 2.2 \times 10^{-16}$ | 89.2% (88–90.6%) | $W = 2776, p = 0.1418$ | 55% (37.6–61.4%) | $W = 906, p = 5 \times 10^{-4}$ | 68.9% (62–76%) |
| | shape | $F_{17,267} = 29.2, p < 2 \times 10^{-16}$ | 91.9% (90.1–93.5%) | $F_{7,134} = 17.4, p = 2 \times 10^{-16}$ | 81.1% (77.2–85.1%) | $F_{5,72} = 12.317, p = 2 \times 10^{-8}$ | 81.4% (75–90%) |
| | form | $F_{12,272} = 46.6, p < 2 \times 10^{-16}$ | 94.8% (93.8–95.9%) | $F_{8,133} = 15.591, p = 4 \times 10^{-16}$ | 81.2% (78–84.3%) | $F_{8,69} = 10.4, p = 8 \times 10^{-8}$ | 82.9% (76–90%) |
| | allometries | $F_{98,184} = 1.41, p = 0.022$ | | $F_{98,41} = 0.84, p = 0.76$ | | $F_{73,2} = 1.06, p = 0.6$ | |
| | size variance | $x^2 = 16.8, p = 4 \times 10^{-5}$ (Dogs > Wolves) | | $x^2 = 0.25, p = 0.62$ | | $x^2 = 3.3, p = 0.07$ | |
| | shape variance | $d_{var} = 0.0001, p = 0.01$ (Dogs > Wolves) | | $d_{var} = 7 \times 10^{-5}, p = 0.281$ | | $d_{var} = 4 \times 10^{-5}, p = 0.6$ | |
| mandibula | size | $W = 12\,156, p < 2 \times 10^{-16}$ | 79% (77.6–80.3%) | $W = 3435, p = 2 \times 10^{-10}$ | 78.5% (76–81.3%) | $W = 655, p = 5 \times 10^{-3}$ | 63.5% (57.9–71%) |
| | shape | $F_{14,232} = 22.79, p < 2 \times 10^{-16}$ | 87.2% (85.1–89.4%) | $F_{6,127} = 12.7, p = 3 \times 10^{-11}$ | 78.6% (74–82.3%) | $F_{6,60} = 10.7, p = 4 \times 10^{-8}$ | 85% (76.3–92.1%) |
| | form | $F_{15,231} = 39.08, p < 2 \times 10^{-16}$ | 93.7% (92.6–95.2%) | $F_{7,126} = 18.104, p < 2 \times 10^{-16}$ | 83.8% (80.2–87.5%) | $F_{5,61} = 12.68, p = 2 \times 10^{-8}$ | 84.2% (78.9–89.5%) |
| | allometries | $F_{26,218} = 4.52, p = 1.8 \times 10^{-10}$ | | $F_{26,105} = 4.85, p = 3 \times 10^{-9}$ | | $F_{26,38} = 1.42, p = 0.15$ | |
| | size variance | $x^2 = 0.04, p = 0.83$ | | $x^2 = 2.94, p = 0.08$ | | $x^2 = 2.68, p = 0.101$ | |
| | shape variance | $d_{var} = 0.0004, p = 0.026$ (Dogs > Wolves) | | $d_{var} = 3 \times 10^{-4}, p = 0.202$ | | $d_{var} = 5 \times 10^{-4}, p = 0.178$ | |

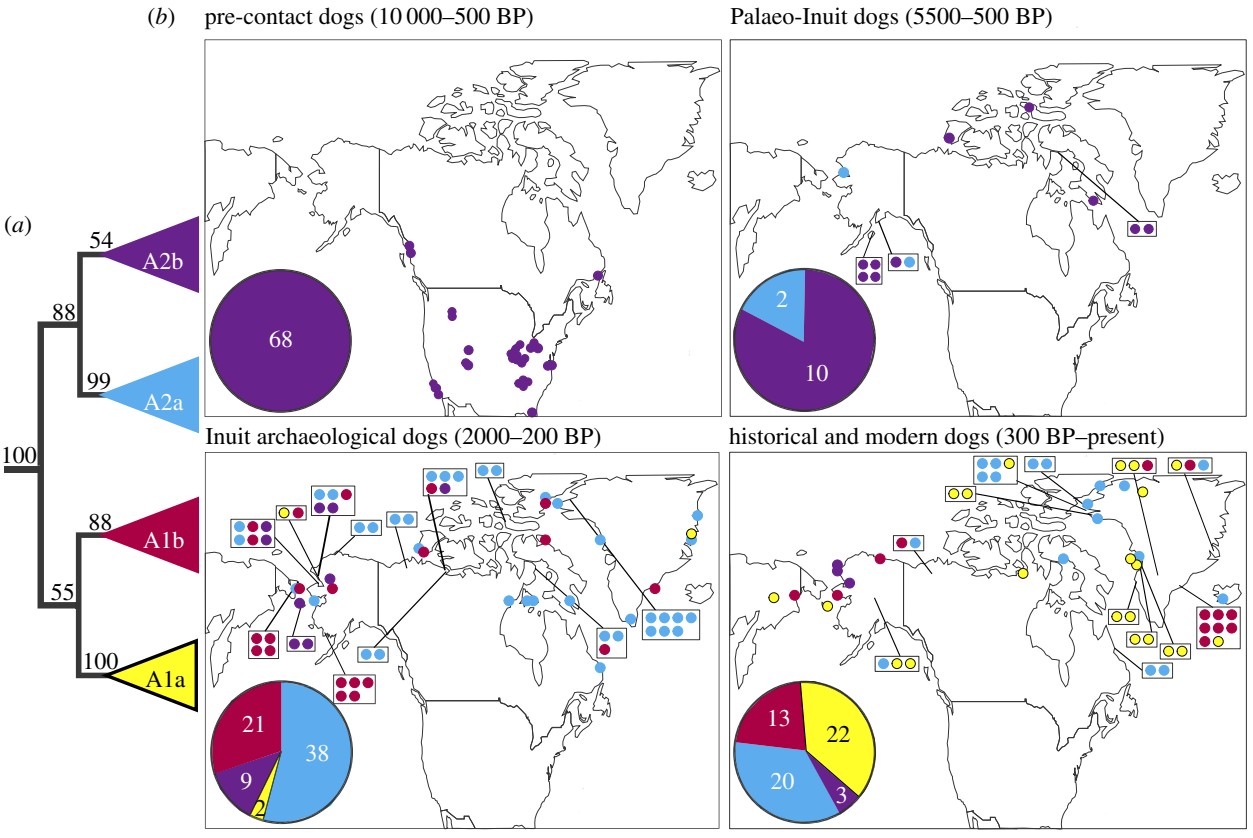

**Figure 2.** Phylogenetic topology and geographic distribution of haplotypes through time. (*a*) The A-clade mitochondrial haplotypes of dogs inferred by maximum-likelihood analyses depicting the four subclades discussed in the text with their respective bootstrap support (for the whole tree see electronic supplementary material). (*b*) Geographical origin of North American dog samples and cultural affiliation. Pie charts indicate the abundance of subclades. Sites with more than one sample are shown in boxes with representation of sample number and haplotype. Modern samples outside of the North American Arctic were excluded from the map and pie chart. Culture dates represent the earliest and latest appearance of each group in the North American Arctic within this dataset [6]. (Online version in colour.)

Inuit dog lineage have been preserved in both modern Canadian Inuit and Greenland Sledge Dogs, the mitochondrial haplotype diversity of the population has changed. The low frequency of the A1a haplotypes in Inuit dog populations (2.9%) stands in stark contrast to the high frequency of Ala haplotypes in recent Arctic dogs (37%). We observed no A1a haplotypes in Palaeo-Inuit dog populations. Our results show that although modern North American Arctic dogs are descended in large part from Inuit period populations there has been regional European A1a introgression during the historical/modern era contributing to the increase in frequency of A1a.

## (b) Canids as tools for the adaptation of humans to Arctic environments

Dogs are an important cultural symbol in the North American Arctic where sledge dogs and dog traction are highly visible components of Arctic identity [33–35]. Dogs were used not only for traction and sledging, but also for hunting, clothing and occasionally as food either preferentially, or during periods of famine [36]. Wolves and other wild canids were additionally exploited for their pelts which protect from frost buildup, or killed as a precaution against potential predation or conflict [37,38].

While most Arctic assemblages primarily consist of domestic canid specimens, there is evidence that wolves were also exploited during the Inuit period. On the basis of size, a large mandible from Nunalleq—a Thule Inuit site in southwestern Alaska [39]—was provisionally identified as a grey wolf (AL2797), alongside a large number of dog specimens

[38,40]. Phylogenetic analysis of the mitochondrial genome from this individual places it among modern and historical wolves from Alaska and Canada (electronic supplementary material, figure S3). This identification was further confirmed by multiple analyses based on low coverage nuclear genome obtained from this individual (electronic supplementary material, figure S10). Two additional Alaskan samples (TRF.02.27, TRF.02.28) taken from clothing made of canid pelts held in ethnographic collections are also likely made from wolf pelts on the basis of their mitochondrial genomes (electronic supplementary material, figure S4) indicating that the use of wolf pelts continued into at least the historical period despite easy access to dogs.

Both nineteenth century Arctic explorers and twentieth century anthropologists reported that Arctic groups often encouraged hybridization of their dogs with wolf populations in order to maintain and strengthen their lineages [41–44]. In the wild, asymmetric bias has been seen with evidence for female wolf–male dog hybridization being dominant and only rare instances of male wolf–female dog hybridization being observed [45]. While the nature of female wolf–male dog hybridization would result in the offspring carrying wolf mitochondrial genomes it also would more than likely exclude these offspring from being in a domestic archaeological context. On the other hand, intentional hybridization of dogs and wolves would likely have been biased towards male wolves mating with female dogs through the deliberate picketing of female dogs in oestrus [46]. The sexual asymmetry in wolf hybridization remains unclear, although explanations for the limited male wolf–female dog hybridization linked to

biological and behavioural constraints, such as male wolf aggression towards dogs, social compatibility and fertility cycles [45,47], mitochondrial introgression from wolves into dogs is improbable, and we did not observed it in our mitochondrial dataset. Anecdotes of hybridization between wolves and dogs are nevertheless widespread in Greenland today, and these reports suggest that hybrids often make poor sledge dogs, and that wolf traits are reportedly selected against [44,46]. This, combined with the observed infrequency of gene flow between wolves and dogs over thousands of years [48,49] makes it unlikely (though not impossible) that any similarities observed between Inuit dogs and Arctic wolves is the result of systematic hybridization, and an analysis of nuclear DNA will clarify this. In addition, the larger cranial and mandible size of the Inuit dogs, when compared with those from the Palaeo-Inuit period, would have been an advantage for their role in transportation and traction during this period though the overall robusticity of the dogs is difficult to detect from skull elements alone. More extensive analyses of postcranial elements could further quantify the unique morphology required for prolonged specialization in sledge pulling.

## (c) Novel X-clade dogs in Eastern Siberia and Alaska

Phylogenetic analysis also revealed seven canid mitogenomes (03.P04.H1.1024, AL2990, AL2991, AL3004, AL3053, AL3007, TRF.02.29) clustering with modern Eurasian wolves, outside any known domestic dog clades, forming a novel subclade referred to in this study as X clade (electronic supplementary material, figures S2, S4–S6 and S8). Three of the samples in X clade are derived from archaeological contexts: two from Neolithic Siberian canids excavated at the Boisman II site in Khasansky District of Primorsky Krai, Russia (AL2990, AL2991; see electronic supplementary material, table S5 for radiocarbon dates) and one dog from the Birnirk site, Pajpel'gak in Chukotka (03.P04.H1.1024). Additionally, this novel subclade contains four historical dogs from Kamchatka, Chukotka, Bering Island and Alaska. While the origins of the fur from the ethnographic Alaskan clothing (TRF.02.29) cannot be conclusively confirmed as dog based on the sample record alone, the ethnographic samples of aboriginal dogs collected from Kamchatka, Chukotka and Bering Island have a definitive classification, having been collected from dogs during the twentieth century (AL3004, AL3053, AL3007; M. Sablin 2019, personal communication).

While the X-clade clusters with modern grey wolves, the inclusion of several recent dogs with known origin eliminates the possibility of this clade representing grey wolves. Although, as previously discussed, interbreeding between dogs and wolves is generally biased towards the mating of male wolves with female dogs which would not result in the passing of mitochondrial genomes from wolves to dogs. The X clade could represent an ancient introgression event of mitochondrial wolf haplotypes into dogs. Additionally, the ancient origin of the clade is demonstrated by sample AL2991 (BOIS 5), which was directly dated to between 6660 and 6495 cal. BP, falling within the clade. This ancient origin is corroborated by Bayesian tree, generated with BEAST2.4 and calibrated with dated samples, estimating the origin of the clade between 5300 and 10 000 years BP (electronic supplementary material, figure S8). Due to constraints on coverage used in the Bayesian analysis sample AL2991, the oldest dated sample in the clade, was not included in the analysis. The inclusion of AL2991 may have pushed the

age of the clade backwards as AL2991 lies well within rather than basal to the clade and is a few millenia older than the samples included in the Bayesian analysis. Even AL2991 included in the Bayesian analysis the sample age falls within the estimated window of origin for the X clade. The comparatively recent origin of this clade, compared to some of the more prominent mitochondrial clades, suggests that the X clade was not present in dogs at the time of domestication. The X clade lineage also appears in historical dogs, demonstrating the continuation of the lineage until at least 75 years ago. The single individual carrying an X clade mitochondrial genome in Alaska may indicate a relatively recent introduction of the lineage to the North American Arctic after the arrival of the Inuit, explaining why the lineage did not spread across the North American Arctic during the Inuit migration and expansion like A1a and A1b haplotypes. The absence of the X-clade haplotypes in published studies speaks to the apparent low frequency and restricted distribution of these haplotypes to far eastern Siberia and Alaska. Furthermore, the absence of these haplotypes in modern dogs may reflect the lack of systematic sampling in the region to date, the low frequency or the disappearance of these haplotypes.

## 3. Conclusion

The phenotypic and genetic data presented here suggest that a novel dog population that was morphologically divergent from, and genetically more diverse than earlier Palaeo-Inuit dogs, accompanied Inuit migrants into and across the North American Arctic. The Inuit migration represents a significant episode in the history of dogs in the North American Arctic, and the dispersal of Inuit culture is mirrored in the dispersal of its genetically distinct dogs. More specifically, our data indicate that though dogs that possessed A2a signatures were present in the North American Arctic prior to the arrival of the Inuit, they were responsible for dominance and eastward expansion of the A2a subclade. The settlement of the North American Arctic by Inuit cultures brought a more mitochondrially diverse and morphologically distinct dog population, and the subsequent European colonization of the North American Arctic further influenced the mitochondrial diversity in more recent centuries. Despite the fact that dog sledging is widely associated with the North American Arctic today, sledging was probably less common prior to the Inuit period. The preservation of these distinctive Inuit dogs is likely a reflection of the highly specialized role that dogs played in both long-range transportation and daily subsistence practices in Inuit society. The legacy of these Inuit dogs survives today in Arctic sledge dogs, making them some of the last remaining descendant populations of a pre-European dog lineage in the Americas.

## 4. Material and methods

### (a) Ancient and modern specimens

Archaeological and ethnographic materials were sampled from Palaeo-Inuit, Inuit, historical and modern contexts across the North American Arctic, subarctic Eastern Canada, Iceland and eastern Siberia. A total genetic dataset of 628 specimens was composed of 186 novel sequences (table 1) and 221 GenBank entries (electronic supplementary material, table S1 and S2). A total of 512 specimens were analysed using GMM, and the dataset

includes 40 crania, 247 mandibles and 284 lower first molars (59 individual specimens were analysed by more than one element; electronic supplementary material, table S3).

## (b) Ancient DNA extraction and sequencing

All samples were processed in facilities dedicated to ancient DNA analyses and all PCRs were performed in separate facilities [50]. Ancient DNA laboratory work was conducted at four institutions: the Swedish Museum of Natural History, the Centre for GeoGenetics at the Natural History Museum of Denmark, University of Oxford's PalaeoBARN Lab, and the Veterinary Genetics Laboratory (VGL) ancient DNA facility at the University of California, Davis (UC Davis).

Total genomic DNA was extracted from 628 samples from across Russia and Arctic North America using modified DNA extraction methods from previous studies ([51,52], see electronic supplementary material). Complete mitochondrial genomes were obtained through either shotgun sequencing (422 samples) or mitochondrial capture approaches (308 samples). Details of the approaches and library preparation are in the electronic supplementary material, text. Following sequencing, the reads from each sample were mapped to the CanFam3.1 reference genome with BWA aln, and aligned reads with a mapping quality score lower than 30 were filtered out of the resulting bam files [53,54]. Subsequently, consensus sequences were called and the sequences were aligned with a panel of reference sequences.

Complete mitochondrial genomes were obtained for 147 samples that had a minimum of threefold read depth over 80% of the mitochondrial genome, and maximum-likelihood phylogenies with constructed using RAxML [55,56]. Additionally, 94 samples had genomes with 10-fold mean coverage, this was reduced to 76 samples with a minimum mean of 10-fold read depth of at least 80% of sites covered. These 76 mitochondrial genomes were then used for further demographic analyses. More robust phylogenetic trees were constructed from the 10-fold dataset with RAxML and BEAST2.4 [55–57]. Effective population size was inferred from Bayesian Skyline plots generated with BEAST2.4 [55–57]. Additionally, fragmentary mitochondrial sequences were obtained from 40 additional samples using Sanger sequencing, all of which possessed sufficient information to assign the resulting D-loop sequences to specific haplotypes. These data were combined with previously published data from pre-contact dogs from the Americas, modern dogs, and modern wolves acquired from GenBank for phylogenetic and demographic analyses (electronic supplementary material, table S2) [4,26].

## (c) Geometric morphometrics

Geometric morphometric analyses were performed on a total of 571 elements (MNI = 401). Mandible and lower M1 shape were analysed in two-dimensional from photographs (see electronic supplementary material, text for protocol details) digitized with 15 landmarks, and two landmarks and 49 sliding semilandmarks, respectively, using tpsDig2 (electronic supplementary material, figure S14) [58]. Crania were analysed in three-dimensional using models built by photogrammetry [59] in Agisoft PhotoScan (Agisoft LLC, St Petersburg, Russia), with 30 landmarks digitized using morphoDig [60] (electronic supplementary material, figure S14).

Prior to analyses, coordinates were superimposed with a generalized procrustes analysis (GPA) [61,62] using the Procrustes distance criterion for optimizing semilandmarks position, and symmetrizing left and right landmarks for the crania.

Size variation was tested with Wilcoxon test and visualized with boxplots showing the log-transformed centroid size. Shape and form (size + shape) variation were explored with principal component analysis (PCA, electronic supplementary material, figure S15), before the differences were tested with MANOVA and visualized using canonical variate analyses (electronic supplementary material, figure S16) based on a reduced number of PCA scores [28]. Differences between groups were also depicted using neighbour-joining networks based on Mahalanobis distances and visualization of shape changes along the discriminant axis. Cross-validation percentages were calculated following [28] and are reported as the mean and 90% confidence interval of 100 discriminant analyses based on ressampled balanced group size. Allometries (size and shape relationships) were tested using MANCOVA. Analyses of Procrustes variance was performed following [27] for shape and Fligner–Killeen tests for size. All statistical analyses were performed in R v. 3.4.3 [63] primarily with the package Morpho [64].

**Data accessibility.** Mitochondrial sequence alignments have been deposited at the European Nucleotide Archive (ENA) with project no. PRJEB31489.

All datasets are available in the electronic supplementary material files and mitochondrial sequence alignments have been deposited at the European Nucleotide Archive (ENA) with project no. PRJEB31489.

**Authors' contributions.** C.A., T.R.F., S.K.B, K.D., L.F.F., G.L., B.S, C.M.D., and A.E. designed research; C.A., T.R.F., A.L., S.K.B., A.H.-B., Z.L., O.L., M.-H.S.S., J.H. and A.E. performed research; C.A., T.R.F., S.K.B, A.H.-B., M.-H.S.S., Z.L., A.T.L., L.F.F. and A.E. analysed data; and C.A., T.R.F., A.L. S.K.B., L.F.F., G.L. and A.E., wrote the paper with contributions from all authors.

**Competing interests.** We declare we have no competing interests.

**Funding.** The following people and institutions supported this study: Doug Anderson; Mogens Andersen; Peter M. Bowers; Joanne Bird; Stephen Brown; Fawn Carter; Neil Duncan; Max Friesen; Stacey Girling-Christie; Kristian Gregersen; John Hoffecker; Claire Houmard; Susan Kaplan; Brian Kooyman; Reinhardt Møbjerg Kristensen; Genevieve LeMoine; Owen Mason; David Morrison; Georg Nyegaard; Anne Mette Olsvig; Jeff Rasic; Knud Rosenlund; Karen Ryan; Chelsea Smith; Kevin P. Smith; Doug Stenton; Daniel Thorleifsen; Andrew Tremayne; Karen Workman; Bill Workman; the Maxwell Museum of Anthropology; Qanirtuuq Inc., Quinhagak, AK; the Nunalleq Archaeology and Culture Center, and Wolfson College, University of Oxford. This research was supported by the following grants: AHRC (grant no. AH/K006029/1), AHRC-LabEx (grant no. AH/N504543/1), European Research Council grant (grant no. ERC-2013-StG-337574-UNDEAD), Natural Environmental Research Council grants (grant nos. NE/K005243/1, NE/K003259/1 and 2210 GG005 RGA1521), National Science Foundation's Office of Polar Programs (grant nos. NSF-ARC-1108175 and NSF-PLR-1304810), the EU-funded ITN project ArchSci2020 (grant no. 676154), Marie Skłodowska-Curie action WhereWolf (grant no. 655732), the Qimmeq project, the Velux Foundations, the Aage og Johanne Louis-Hansens Fond and the Wellcome Trust (grant no. 210119/Z/18/Z).

**Acknowledgements.** We acknowledge the participation of ZIN RAS (state assignment no. AAAA-A17-11702281019503), and the support from Science for Life Laboratory, the National Genomics Infrastructure, and UPPMAX (project no. b2014314) for providing assistance in massive parallel sequencing and computational infrastructure.

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
