## [Reviewer comments · Proceedings of the Royal Society B: Biological Sciences]

Review History

RSPB-2019-1929.R0 (Original submission)

Review form: Reviewer 1

Recommendation

Accept with minor revision (please list in comments)

Scientific importance: Is the manuscript an original and important contribution to its field?

Excellent

General interest: Is the paper of sufficient general interest?

Good

Quality of the paper: Is the overall quality of the paper suitable?

Good

Is the length of the paper justified?

Yes

Should the paper be seen by a specialist statistical reviewer?

No

Do you have any concerns about statistical analyses in this paper? If so, please specify them explicitly in your report.

Yes

It is a condition of publication that authors make their supporting data, code and materials available - either as supplementary material or hosted in an external repository. Please rate, if applicable, the supporting data on the following criteria.

Is it accessible?

Yes

Is it clear?

N/A

Is it adequate?

N/A

Do you have any ethical concerns with this paper?

No

Comments to the Author

I list some minor suggestions.

p.4 Abstract: the data do not reveal that dogs aided Inuit expansion. Change to "may have" aided

Keywords: should include dogs or *Canis familiaris*, Arctic, or circumpolar

Why is *Canis familiaris* used, and not *Canis lupus familiaris*? Especially considering the discussion of hybridization

p. 5 ...lineage that was introduced (not were)

p. 9 ...data show (not shows)

p. 11 ...likely have been biased

p. 12 calibration (not calibrate)

...Inuit culture is mirrored in the dispersal of its (not their)

Inuit dogs may be a reflection (not is. As per Abstract, the study does not directly address this)

Also, the authors may consider using "specimens" instead of "remains" when referring to preserved archaeological dog specimens (remains is frequently used as a verb in the text)

Review form: Reviewer 2

Recommendation

Accept with minor revision (please list in comments)

Scientific importance: Is the manuscript an original and important contribution to its field?

Excellent

General interest: Is the paper of sufficient general interest?

Good

Quality of the paper: Is the overall quality of the paper suitable?

Excellent

Is the length of the paper justified?

Yes

Should the paper be seen by a specialist statistical reviewer?

No

Do you have any concerns about statistical analyses in this paper? If so, please specify them explicitly in your report.

No

It is a condition of publication that authors make their supporting data, code and materials available - either as supplementary material or hosted in an external repository. Please rate, if applicable, the supporting data on the following criteria.

Is it accessible?

Yes

Is it clear?

Yes

Is it adequate?

Yes

Do you have any ethical concerns with this paper?

No

Comments to the Author

This well-written manuscript by a large and highly regarded collaborative group will be a welcome addition to the growing literature on the evolution of diversity in dogs. Its strength lies in the substantial sample sizes employed to use ancient dogs as proxies to address questions relating to human dispersal in the Western Hemisphere, inferences regarding the subsistence strategies of these early prehistoric peoples of the arctic, and the joint use of both morphometric and molecular data to document patterns of variation in prehistoric dogs.

Unlike many such analyses on ancient remains, the sample sizes are substantial in both the morphometric and molecular analyses, adding confidence to the inferences reached. The analytical methods are appropriate (although I am not expert in the geometric morphometric analyses), and inferences consistent with results. The identification of the novel X-clade is of considerable import. It demonstrates that there is diversity in canid lineages that has yet to be

observed, indicating, as the authors note, the importance of sampling in such studies. This lineage, and others yet to be identified, will likely spur additional work in the future. In general, I found this a most interesting paper. It will be of interest to students of domestication, arctic prehistory and adaptation, and those interested in human dispersals into the North American arctic.

I have few substantive suggestions for change. However, I do have minor comments on Table 2 and Figure 2. In the copy I reviewed Table 2 is very condensed, in a small font, and is very 'cramped' and difficult to read. Placing it in landscape rather than portrait orientation would be helpful to readers. This may, of course, have been the expectation for the final published version. It is packed with important information regarding the analyses and it would be unfortunate if readers could not easily access it.

Figure 2 is a very informative illustration, and captures much of the sample structure and analytical results effectively. However, it would be somewhat clearer, or easier to match the tree with the four panels if the tree (Fig. 2a) were rotated on its axes so that the A2a and A2b subclades were on top since those are the only subclades illustrated in the top panels of Fig. 2b. Similarly, placing the A1a and A1b lineages on the bottom of the tree would provide a better match for the bottom panels where their frequencies are actually illustrated.

At the top of page 2 of the manuscript, line 2, I believe it should read 'Aside from...', not 'Aside for...'

Decision letter (RSPB-2019-1929.R0)

02-Oct-2019

Dear Dr Evin:

Your manuscript has now been peer reviewed and the reviews have been assessed by an Associate Editor. The reviewers' comments (not including confidential comments to the Editor) and the comments from the Associate Editor are included at the end of this email for your reference. As you will see, the reviewers and the Editors have raised some concerns with your manuscript and we would like to invite you to revise your manuscript to address them.

When submitting your revision please upload a file under "Response to Referees" - in the "File Upload" section. This should document, point by point, how you have responded to the reviewers' and Editors' comments, and the adjustments you have made to the manuscript. We

require a copy of the manuscript with revisions made since the previous version marked as 'tracked changes' to be included in the 'response to referees' document.

Research ethics:

Use of animals and field studies:

If you wish to submit your data to Dryad (<http://datadryad.org/>) and have not already done so you can submit your data via this link [http://datadryad.org/submit?journalID=RSPB&manu=\(Document not available\)](http://datadryad.org/submit?journalID=RSPB&manu=(Document%20not%20available)), which will take you to your unique entry in the Dryad repository.

Online supplementary material will also carry the title and description provided during submission, so please ensure these are accurate and informative. Note that the Royal Society will

not edit or typeset supplementary material and it will be hosted as provided. Please ensure that the supplementary material includes the paper details (authors, title, journal name, article DOI). Your article DOI will be 10.1098/rspb.[paper ID in form xxxx.xxxx e.g. 10.1098/rspb.2016.0049].

Please submit a copy of your revised paper within three weeks. If we do not hear from you within this time your manuscript will be rejected. If you are unable to meet this deadline please let us know as soon as possible, as we may be able to grant a short extension.

Best wishes,

Dr Daniel Costa
mailto:proceedingsb@royalsociety.org

Associate Editor
Board Member: 1
Comments to Author:

Two referees have evaluated your manuscript and provide positive comments. They do, however, highlight some few issues that need to be taken into consideration. In particular, it is important that you improve Table 2 and Figure 2 so as to make them easier to interpret.

Reviewer(s)' Comments to Author:

Referee: 1

Comments to the Author(s)

I list some minor suggestions.

p.4 Abstract: the data do not reveal that dogs aided Inuit expansion. Change to "may have" aided

Keywords: should include dogs or *Canis familiaris*, Arctic, or circumpolar

Why is *Canis familiaris* used, and not *Canis lupus familiaris*? Especially considering the discussion of hybridization

p. 5 ...lineage that was introduced (not were)

p. 9 ...data show (not shows)

p. 11 ...likely have been biased

p. 12 calibration (not calibrate)

...Inuit culture is mirrored in the dispersal of its (not their)

Inuit dogs may be a reflection (not is. As per Abstract, the study does not directly address this)

Also, the authors may consider using "specimens" instead of "remains" when referring to preserved archaeological dog specimens (remains is frequently used as a verb in the text)

Referee: 2

Comments to the Author(s)

This well-written manuscript by a large and highly regarded collaborative group will be a welcome addition to the growing literature on the evolution of diversity in dogs. Its strength lies in the substantial sample sizes employed to use ancient dogs as proxies to address questions relating to human dispersal in the Western Hemisphere, inferences regarding the subsistence

strategies of these early prehistoric peoples of the arctic, and the joint use of both morphometric and molecular data to document patterns of variation in prehistoric dogs.

Unlike many such analyses on ancient remains, the sample sizes are substantial in both the morphometric and molecular analyses, adding confidence to the inferences reached. The analytical methods are appropriate (although I am not expert in the geometric morphometric analyses), and inferences consistent with results. The identification of the novel X-clade is of considerable import. It demonstrates that there is diversity in canid lineages that has yet to be observed, indicating, as the authors note, the importance of sampling in such studies. This lineage, and others yet to be identified, will likely spur additional work in the future. In general, I found this a most interesting paper. It will be of interest to students of domestication, arctic prehistory and adaptation, and those interested in human dispersals into the North American arctic.

I have few substantive suggestions for change. However, I do have minor comments on Table 2 and Figure 2. In the copy I reviewed Table 2 is very condensed, in a small font, and is very 'cramped' and difficult to read. Placing it in landscape rather than portrait orientation would be helpful to readers. This may, of course, have been the expectation for the final published version. It is packed with important information regarding the analyses and it would be unfortunate if readers could not easily access it.

Figure 2 is a very informative illustration, and captures much of the sample structure and analytical results effectively. However, it would be somewhat clearer, or easier to match the tree with the four panels if the tree (Fig. 2a) were rotated on its axes so that the A2a and A2b subclades were on top since those are the only subclades illustrated in the top panels of Fig. 2b. Similarly, placing the A1a and A1b lineages on the bottom of the tree would provide a better match for the bottom panels where their frequencies are actually illustrated.

At the top of page 2 of the manuscript, line 2, I believe it should read 'Aside from...', not 'Aside for...'

Author's Response to Decision Letter for (RSPB-2019-1929.R0)

See Appendix A.

Decision letter (RSPB-2019-1929.R2)

28-Oct-2019

Dear Dr Evin

I am pleased to inform you that your Review manuscript RSPB-2019-1929.R1 entitled "Specialised sledge dogs accompanied Inuit dispersal across the North American Arctic" has been accepted for publication in Proceedings B.

The referee(s) do not recommend any further changes. Therefore, please proof-read your manuscript carefully and upload your final files for publication. Because the schedule for publication is very tight, it is a condition of publication that you submit the revised version of your manuscript within 7 days. If you do not think you will be able to meet this date please let me know immediately.

To upload your manuscript, log into <http://mc.manuscriptcentral.com/prsb> and enter your Author Centre, where you will find your manuscript title listed under "Manuscripts with Decisions." Under "Actions," click on "Create a Revision." Your manuscript number has been appended to denote a revision.

You will be unable to make your revisions on the originally submitted version of the manuscript. Instead, upload a new version through your Author Centre.

1) A text file of the manuscript (doc, txt, rtf or tex), including the references, tables (including captions) and figure captions. Please remove any tracked changes from the text before submission. PDF files are not an accepted format for the "Main Document".

2) A separate electronic file of each figure (tiff, EPS or print-quality PDF preferred). The format should be produced directly from original creation package, or original software format. Please note that PowerPoint files are not accepted.

3) Electronic supplementary material: this should be contained in a separate file from the main text and the file name should contain the author's name and journal name, e.g. `authorname_procb_ESM_figures.pdf`

All supplementary materials accompanying an accepted article will be treated as in their final form. They will be published alongside the paper on the journal website and posted on the online figshare repository. Files on figshare will be made available approximately one week before the accompanying article so that the supplementary material can be attributed a unique DOI. Please see: <https://royalsociety.org/journals/authors/author-guidelines/>

4) Data-Sharing and data citation

It is a condition of publication that data supporting your paper are made available. Data should be made available either in the electronic supplementary material or through an appropriate repository. Details of how to access data should be included in your paper. Please see <https://royalsociety.org/journals/ethics-policies/data-sharing-mining/> for more details.

<http://datadryad.org/submit?journalID=RSPB&manu=RSPB-2019-1929.R1> which will take you to your unique entry in the Dryad repository.

Once again, thank you for submitting your manuscript to Proceedings B and I look forward to receiving your final version. If you have any questions at all, please do not hesitate to get in touch.

Sincerely,
Dr Daniel Costa
Editor, Proceedings B
<mailto:proceedingsb@royalsociety.org>

Associate Editor Board Member

Comments to Author:

The revised version of the manuscript incorporates all suggestions made by the referees. There are some few problems with the English (missing articles -a, the, and ", ". Please go over the manuscript very carefully and correct these mistakes.

Decision letter (RSPB-2019-1929.R2)

05-Nov-2019

Dear Dr Ameen

I am pleased to inform you that your manuscript entitled "Specialised sledge dogs accompanied Inuit dispersal across the North American Arctic" has been accepted for publication in Proceedings B.

Your article has been estimated as being 9 pages long. Our Production Office will be able to confirm the exact length at proof stage.

Open Access

Paper charges

Sincerely,

Proceedings B
mailto: proceedingsb@royalsociety.org

Appendix A

Institut des Sciences de l'Evolution - Montpellier
CNRS, Université de Montpellier, IRD, EPHE
2 place Eugène Bataillon, CC065
34095 Montpellier, Cedex 5, France

Email: allowen.evin@umontpellier.fr

23th October 2019

Dear Dr. Costa,

We would like to submit a revision of our manuscript entitled '*Specialised sledge dogs accompanied Inuit dispersal across the North American Arctic*'.

We have agreed with all reviewer's comments and the comments from the Associate Editor. Please find below a point by point list of how we have responded to the reviewers' and Editors' comments, and the adjustments we have made to the manuscript.

We are looking forward to your consideration of our revised manuscript.

Yours sincerely,
Allowen EVIN

Please find below our response to the referees comments. Our answered are shaded in grey and modified text highlighted in yellow. Below our response we have included changes not requested by the referees but that we thought to be important to add before publication.

Board Member: 1

Comments to Author:

Two referees have evaluated your manuscript and provide positive comments. They do, however, highlight some few issues that need to be taken into consideration. In particular, it is important that you improve Table 2 and Figure 2 so as to make them easier to interpret.

Reviewer(s)' Comments to Author:

Referee: 1

Comments to the Author(s)

I list some minor suggestions.

p.4 Abstract: the data do not reveal that dogs aided Inuit expansion. Change to "may have" aided

The text has been modified and appear now as bellow:

Together, our data reveal that Inuit dogs derive from a secondary pre-contact migration of dogs distinct from Paleo-Inuit dogs, and **most likely aided** the Inuit expansion across the North American Arctic beginning around 1,000 BP. We also identified a novel mitochondrial clade in eastern Siberia and Alaska.

Keywords: should include dogs or *Canis familiaris*, Arctic, or circumpolar

The keyword list has been modified and include now both *Canis lupus familiaris* and circumpolar:

Archaeology; Geometric morphometrics; Ancient DNA; Migration; ***Canis lupus familiaris***; **Circumpolar**

Why is *Canis familiaris* used, and not *Canis lupus familiaris*? Especially considering the discussion of hybridization

The text has been modified accordingly

p. 5 ...lineage that was introduced (not were)

Corrected

p. 9 ...data show (not shows)

Corrected

p. 11 ...likely have been biased

Corrected

p. 12 calibration (not calibrate)

Corrected

...Inuit culture is mirrored in the dispersal of its (not their)

Corrected

Inuit dogs may be a reflection (not is. As per Abstract, the study does not directly address this)

The text has been modified accordingly

Also, the authors may consider using "specimens" instead of "remains" when referring to preserved archaeological dog specimens (remains is frequently used as a verb in the text)

The text has been modified accordingly

Referee: 2

Comments to the Author(s)

This well-written manuscript by a large and highly regarded collaborative group will be a welcome addition to the growing literature on the evolution of diversity in dogs. Its strength lies in the substantial sample sizes employed to use ancient dogs as proxies to address questions relating to human dispersal in the Western Hemisphere, inferences regarding the subsistence strategies of these early prehistoric peoples of the arctic, and the joint use of both morphometric and molecular data to document patterns of variation in prehistoric dogs.

Unlike many such analyses on ancient remains, the sample sizes are substantial in both the morphometric and molecular analyses, adding confidence to the inferences reached. The analytical methods are appropriate (although I am not expert in the geometric morphometric analyses), and inferences consistent with results. The identification of the novel X-clade is of considerable import. It demonstrates that there is diversity in canid lineages that has yet to be observed, indicating, as the authors note, the importance of sampling in such studies. This lineage, and others yet to be identified, will likely spur additional work in the future. In general, I found this a most interesting paper. It will be of interest to students of domestication, arctic prehistory and adaptation, and those interested in human dispersals into the North American arctic.

I have few substantive suggestions for change. However, I do have minor comments on Table 2 and Figure 2. In the copy I reviewed Table 2 is very condensed, in a small font, and is very 'cramped' and difficult to read. Placing it in landscape rather than portrait orientation would be helpful to readers. This may, of course, have been the expectation for the final published version. It is packed with important information regarding the analyses and it would be unfortunate if readers could not easily access it.

We agree that Table 2 was to packed in the submitted version. We have now submitted the table as an independent file. We have decided to keep the table as it was because it contains important information concerning the statistical tests.

Figure 2 is a very informative illustration, and captures much of the sample structure and analytical results effectively. However, it would be somewhat clearer, or easier to match the tree with the four panels if the tree (Fig. 2a) were rotated on its axes so that the A2a and A2b subclades were on top since those are the only subclades illustrated in the top panels of Fig. 2b. Similarly, placing the A1a and A1b lineages on the bottom of the tree would provide a better match for the bottom panels where their frequencies are actually illustrated.

Figure 2 has been modified accordingly.

At the top of page 2 of the manuscript, line 2, I believe it should read 'Aside from...', not 'Aside for...'

Corrected

Changes not requested by the referees:

- We have removed an extra 'the' to the title.

Specialised sledge dogs accompanied (the) Inuit dispersal across the North American Arctic

- We have modified the following sentence to gain in precision (p7):

This ancient origin is corroborated by bayesian tree, generated with BEAST2.4 and calibration with dated samples, estimating the origin of the clade between 5,300 and 10,000 years BP (Fig. S8)

Before it was:

This ancient origin is corroborated by bayesian tree, generated with BEAST2.4 and calibrate with dated samples, estimating the origin of the clade around-5,300 years BP (Fig. S8).

- We have modified the following sentence to gain in precision (p7)

The inclusion of AL2991 may have pushed the age of the clade backwards, as AL2991 lies well within rather than basal to the clade, although regardless the age of the sample it falls within the estimated window of origin for the X-clade.

Before it was:

The inclusion of AL2991 would likely push the age of the clade backwards as AL2991 lies well within rather than basal to the clade.

- We have extended the legends of EMS figures 7 and 8:

Fig. S7: Bayesian mitochondrial phylogeny constructed with BEAST2.4 for A-clade with dated nodes from samples with a minimum mean of 10x read depth. The subclades of the A-clade have been coloured according with Fig. 2, purple for clade A2b, blue for clade A2a, red for clade A1b, and yellow for clade A1a.

Fig. S8: Bayesian mitochondrial phylogeny constructed with BEAST2.4 for whole dataset with dated nodes from samples with a minimum mean of 10x read depth. The subclades of the A-clade have been coloured according with Fig. 2, purple for clade A2b, blue for clade A2a, red for clade A1b, and yellow for clade A1a.

- We have also made some minor, mainly grammatical, changes to gain in clarity (they have been kept in track changes in the modified version of the manuscript).
- We made changes in table 2: we replaced the size variance test to a more appropriate approach and rerun all the analyses with updated versions of the functions. The text has been modified accordingly. While the numbers changed, overall trends observed in the data remained the same.